# Can Collecting Water Fees Really Promote Agricultural Water-Saving? Evidence from Seasonal Water Shortage Areas in South China

**Xuerong Li**

Water Economics and Water Rights Research Center, Nanchang Institute of Technology, Nanchang 330099, China; xuerong_li@nit.edu.cn or 2017994632@nit.edu.cn

**Abstract:** Under the influence of the extreme climate, South China frequently experiences a seasonally arid climate, resulting in seasonal water shortages, and threatening local food and water security. To cope with climate change, agricultural water-saving is inevitable. However, compared with the North, South China is rich in water resources, farmers' water-saving awareness is weak, and most areas do not charge water fees, so it is difficult to promote agricultural water-saving; therefore, farmers' agricultural water-saving behavior is worth discussing. Based on a survey and empirical analysis, this study identifies the key determinants of farmers' agricultural water-saving behavior, particularly to verify whether collecting water fees helps to promote agricultural water-saving. A structured questionnaire was administered to a random sample of 660 farmers in South China with seasonal water shortage. A binary logistic regression model was used to examine the determinants. The results revealed that 15.30% (101) of farmers paid agricultural water fees, 26.97% (178) of farmers had agricultural water-saving behavior, and among these, 43.82% (78) of farmers paid agricultural water fees. The results indicated that water fee collecting, water resource dependence, agricultural water service satisfaction, and water-saving policy publicity positively and significantly influenced farmers' agricultural water-saving behavior, while farm size and age of household head showed a negative influence. Results also revealed that collecting water fees can indeed promote agricultural water-saving in seasonal water shortage areas of South China. This study recommends that policy makers take measures to improve agricultural water charges policies, strengthen irrigation services, and increase the publicity of agricultural water-saving policies.

**Keywords:** agricultural water management; agricultural water-saving; water fees; seasonal water shortage; climate change

## 1. Introduction

China is confronted with a severe shortage of water resources [1]. The annual water availability per capital is approximately one-quarter of the world average [2]. With the rapid development of economy and society, the contradiction between supply and demand of water resources has become increasingly prominent. Coupled with the frequent occurrence of extreme climate, this shortage has become the bottleneck of ecological civilization construction, economy, and society sustainable development [3]. For this reason, the Chinese government put forward a new strategic plan, which is "implementing the national water-saving action," indicating that water-saving has become the will of the state and the action of the entire nation. Agriculture is a significant user of water, and also has significant potential in terms of water-saving. In 2021, China's agricultural water consumption was 364.43 billion m$^3$, accounting for 61.5% of its total water consumption [4]. Agriculture is a big water user, which not only consumes large amounts of water, but also has low efficiency. The use efficiency of China's agricultural irrigation water is 0.568 [4], with a difference of 0.2–0.3 compared with 0.7–0.8 for developed countries [1]. It is imperative to save water and increase efficiency in agriculture. However, China's water resources are

unevenly distributed in time and space. Compared with the North, the South is relatively rich in water resources. But significant changes have taken place in water structure due to population growth and rapid economic and social development, and the demand for water has significantly increased. In addition, frequent extreme climate events have also resulted in seasonal droughts and water shortages, which restrict the sustainable development of economy and society. Since late July 2019, South China has experienced extreme drought conditions (see Figure 1). Most of the seven provinces in this region have experienced 20–80% less precipitation than the same period of the previous year, with an average precipitation of only 204.9 mm, nearly 50% less, the lowest for the period since records began in 1961, with the middle east of Jiangxi Province experiencing 80% less precipitation [5]. In 2022, South China again encountered continuous high temperature and drought, which has become the biggest threat to the autumn harvest. Until September 20, the Central Meteorological Observatory has issued a drought warning for the 34th consecutive day. Climate change causes seasonal water shortages in the South, which restricts economy and society sustainable development. It is necessary to accelerate the promotion of agricultural water-saving in this region, and to improve agricultural water resource use efficiency.

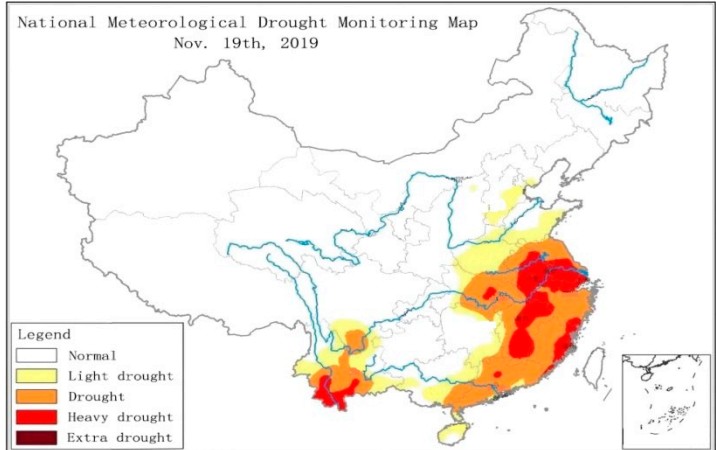

**Figure 1.** The National Meteorological Drought Monitoring Map. Note: The above map was quoted from China National Climate Center.

　　Facing the new situation of agricultural water-saving in the new era, how to leverage economic instruments to promote agricultural water-saving and how to effectively play the decisive role of market allocation of water resources are the key and difficult points of the current agricultural water-saving work. Collecting agricultural water fees has been proved to be an effective means of promoting agricultural water-saving in water deficient areas of Northern China [6]. However, in South China, due to the rich water resources and the lack of measurement facilities, agricultural water fees are charged by area, farmers generally lack water commodity awareness, the concept of water payment and the awareness of water-saving is weak, and water resource waste is extensive. Furthermore, since 2003, the Chinese government has started to subsidize the agricultural sector and pay more attention to farmers' incomes [7]. With the implementation of policies such as agricultural tax relief and the state's granting of farm subsidies, some regions have not collected water fees from farmers, but paid by local finance through transfer payment, which is not conducive to mobilizing farmers' enthusiasm regarding agricultural water-saving. Therefore, China is accelerating the comprehensive reform of agricultural water pricing through the improvement of the agricultural water price formation mechanism. In addition, by establishing precise subsidies and agricultural water-saving reward mechanisms, it aims to give full play to the leverage of price on agricultural water-saving and realize the efficiency of agricultural water-saving. Agricultural water price adjustment and fees collection are the important elements of this round of water price reform. To enhance farmers' water

commodity awareness and stimulate endogenous water-saving power, reforms include the method of collecting agricultural water fees according to the quantity (square) of water. After the reform of agricultural water price, some changes have taken place in farmers' water use behavior. According to the field survey, to cope with the seasonal water shortage caused by climate change, some farmers in Southern China areas also attempted to save water, such as reducing irrigation quantities and times, adjusting crops' planting structure, carrying out agricultural water-saving improvement, and adopting agricultural water-saving technology. For these farmers, what are the factors that affect their agricultural water-saving behavior? Do water fees significantly affect their agricultural water-saving behavior? Can collecting water fees really promote agricultural water-saving? Is there a significant difference in water use behavior between farmers who pay agricultural water fees and those who do not? To explore the answers to these questions is of great practical significance and value for effectively exerting the leverage role of price in agricultural water-saving, promoting agricultural water-saving, and realizing agricultural water-saving efficiency in South China.

Climate change has led to an increase in global average annual temperatures, changes in regional precipitation patterns, changes in river characteristics, and frequent occurrence of extreme climatic events, which pose significant challenges to the sustainable development of agriculture and modern agricultural water management around the world, especially in the arid and semi-arid regions [8]. Measures to achieve the efficient and sustainable use of water resources have become an important issue in water resource management. In terms of the factors that affect a farmer's water-saving behavior, scholars have conducted valuable research [9–28]. Since the mid-1980s, some scholars have investigated the adoption behavior of agricultural water-saving technology in arid areas of the United States, Israel, and other countries; the results indicate that farmers' adoption of water-saving technology is closely related to the water-saving degree, agricultural water price, farm size, agricultural income, water resource tax, water-saving cost, water market, drought severity, availability of irrigation water, government subsidies, water-saving services, and so on [9–22]. Caswell et al. studied the factors influencing the choice of irrigation mode for fruit farmers in California, and the results show that the choice is related to the degree of water-saving, water price, farmers' income level, and water resource use tax [9]. Carey et al. used the stochastic dynamic model of irrigation technology, taking the randomness of future drought degree and the uncertainty of economic incentive factors (such as water price and water market) into account, and concluded that the potential water market makes it easier for farmers with sufficient water resources to adopt water-saving technology than those in areas with a shortage of water resources [10]. Schuck et al. investigated how the degree of drought affected farmers' choice of water-saving irrigation technology, and found that the increase in the degree of drought promoted farmers to adopt more effective sprinkler irrigation technology [11]. Dridi et al. and Dinar et al. found that water price, crop income, government subsidies, availability of water supply, and other factors have a significant impact on water-saving technology [12,13]. Bjornlund et al. revealed that stable water supply in arid areas, increased production and efficiency, and saving costs promote farmers to adopt water-saving irrigation technology [14]. Alam found that longer planting experience, better education level, improved power infrastructure, and awareness of coping with climate change can promote farmers to adopt water-saving irrigation technology [15]. Abdulai et al. argued that age, education, planting patterns, and other factors can significantly affect farmers' adoption of water-saving technology [16]. Cremades et al. suggested that fund subsidies and promotion service policies had an important impact on the adoption of such technology [17]. Rum et al. analyzed the Serbian farmers' economic incentive to use water-saving systems, and the results showed that increased taxes on water may provide incentives for farmers to use water-saving systems [18]. Rozakis used a nonlinear optimization model to examine the impacts of water pricing and CAP reform on water use, irrigation technology use, and farm returns in the region of Thessaly, Greece; the results indicated that water use is more sensitive to water pricing [19]. Qu et al. and Ma

C et al. found that direct subsidy is more efficient comparing to indirect subsidy toward agricultural water-saving, and they suggested that the explicit subsidy of agricultural water prices should be made to save irrigation water [20,21]. Bos and Wolters measured irrigation efficiencies with characteristics of charges and irrigation systems, and the research results revealed that water charges only influence irrigation efficiency if they are levied by volume [22]. In addition, some scholars have investigated Chinese farmers' agricultural water-saving technology adoption behavior. The results indicated that age, education degree, farmers' risk awareness, main family income source, proportion of agricultural income, non-agricultural employment, planting structure, farm size, land fragmentation, water shortage, water infrastructure, water price, technical guidance, extension services, water-saving publicity, government subsidies, and so on have a significant impact on farmers' water-saving technology adoption [23–28]. Using the data of household inspection in nine provinces, Liu et al. conducted an empirical analysis and found that government support, farmers' education degree, water resource shortage level, and farmland fragmentation level have a significant impact on farmers' adoption of water-saving technology [23]. Liu et al. demonstrated that the degree of water shortage and policy support had a significant positive impact on farmers' use of such technology [24]. Xu et al. used a binary logistic model to conduct an empirical analysis on influencing factors of farmers' water-saving technology choices and found that farmers' awareness level regarding water-saving technology, the proportion of agricultural income, household income sources, land acreage, and water-saving policy are important factors to promote water-saving techniques [25]. Based on cross-sectional data of 357 farmers in the Guanzhong Plain, North China, Tang et al. analyzed the adoption of farm-based irrigation water-saving techniques. The results indicated that awareness of water scarcity and financial status enhance the adoption of more advanced techniques [26]. Zhang et al. analyzed new agricultural management entities' adoption behavior of water-saving technology and found that technical guidance, financial subsidies, and cognitive level have significant positive impacts on their adoption behaviors [27]. Gong et al. applied an optimal scaling regression model to analyze the factors affecting farmers' water-saving behaviors. They found that education level, farmland location, proportion of crop income, proportion of water cost to total production cost, and farmers' attitudes toward new technology have significant positive influences on farmers' water-saving behavior [28].

The existing research results are helpful to guide the popularization and application of agricultural water-saving technology, but there is still room for expansion. First, in terms of research content, the existing research focuses on water-saving technology adoption behavior and affecting factors. While water-saving technology adoption is only one aspect of water-saving behavior, which obviously cannot fully reflect the situation of farmers' agricultural water-saving behavior, there is a lack of research on such behavior. Second, in terms of research objects, most of the previous studies on the aforementioned behavior have focused largely on drought areas in North China; there is a lack of research on farmers in seasonal water shortage areas of South China. Under the condition of climate change, these areas also need to implement agricultural water-saving, and the existing research results cannot be fully used to guide the agricultural water-saving practice in southern seasonal water shortage areas. Furthermore, the existing research lacks the theoretical analysis of the mechanism of farmers' water-saving behavior, and it does not introduce the variable of water fees to explore the influencing factors of farmers' water-saving behavior. Therefore, based on the investigation, this study uses a binary logistic model to examine farmers' water use behavior, discusses the factors that affect farmers' agricultural water-saving behavior, and focuses on verifying whether collecting water fees can really promote agricultural water-saving. This is of great practical value to explore the agricultural water-saving behavior of farmers and its influencing factors, which will be conducive to promoting the agricultural water-saving work and continuously improving the utilization efficiency and efficiency of water resources.

This study provides an overview and information useful for agricultural water-saving in seasonal water shortage areas of South China, which helps to realize the efficient and sustainable utilization of water resources. This study aims to fill the gap in the literature by providing information about factors significantly influencing farmers' agricultural water-saving behavior in South China, in particular, verifying whether collecting water fees can promote agricultural water-saving. The study also provides valuable information for agricultural water price reform and agricultural water-saving policy decisions.

## 2. Theoretical Analyses and Hypotheses

### 2.1. Theoretical Analyses

As the basic unit of agricultural production, farmers are the providers and producers of agricultural products, and the consumers of various production factors (such as water resources), and decision makers regarding production. With the development of the social economy, especially rural reform and agricultural modernization, rural production has long been separated from the self-sufficient business model. As an individual of socialized management, farmers are "rational economic individuals". When they make production decisions, they are faced with various internal and external constraints, and will compare the opportunity cost they pay and the maximum profit they obtain when they choose a certain production decision, and pursue the maximum profit as their goal.

Water resource is an essential factor of agricultural production. Based on the goal of profit maximization, farmers will fully consider various factors that affect the input cost of these factors, and decide on the reasonable number of factors to input. The goal of farmers in agricultural production is to maximize the economic profit, and the profit is equal to the total revenue minus the total cost, as expressed in Equation (1):

$$\text{Profit } (\pi) = \text{Total revenue} - \text{Total cost} = TR - TC = P \times Q - C \tag{1}$$

where P is the selling price of agricultural products; Q is the output of agricultural products; and C is the input cost of all production factors, including land, labor, capital, technology, fertilizer, pesticide, water, and other production factors.

For water resources, the important factor of production, to maximize economic profits, the overall goal of farmers is to minimize the cost of water use when P, Q, and the cost of other production factors are fixed. At this time, farmers' water use (input) amount is closely related to water price, water resource endowment, water-saving technology, water resource management policy (e.g., water fees charge or not), and so on. This study mainly analyzes the impact on farmers' agricultural water-saving behavior from the perspective of water price and water fees.

**Case 1.** *Farmers do not need to pay water fees.*

In this case, as farmers do not need to pay water fees and use water for free, the cost of water use is zero. The amount of water consumption does not affect their profit maximization goal, so most farmers tend not to save water.

**Case 2.** *Farmers pay water fees, but the water price is low, and the water fees are not related to the water consumption amount.*

In this case, water fees are a production cost, but as the water price is low, and the water fees are not related to consumption amount (fees are charged by farmland area), fees are a fixed value, and the proportion of water fees in the production cost is low. At this time, the farmer needs to pay a fixed water fee no matter how much water they consume, and they have no incentive to save water, and even tend to use more water as they have paid the water fees.

**Case 3.** *Farmers pay water fees, but the water price is fixed, and the fee is related to the water consumption amount.*

In this case, fees are related to the water consumption amount and as the Chinese government implements total amount control and quota management for agricultural water consumption, the low price (still fixed price) within the quota and the progressive price increase system are implemented if the quota is exceeded (the water price is positively related to the water consumption amount); hence, the water cost for farmers is no longer a fixed value. To maximize profits, farmers must fully consider the water price and reduce the water cost as much as possible. At this time, the agricultural water consumption amount (expressed in $Q_a$) is the demand amount for water resources under various conditions. The demand amount is affected by various factors and is a function of water price and other variables, as expressed in Equation (2):

$$Q_a = Q(P_a, P, T, E, C, G \cdots) \tag{2}$$

where $P_a$ is agricultural water price; P is the price of agricultural products; T is water-saving technology or facilities; E is the scarcity degree of water resources (water resource endowment); C is the water-saving awareness of farmers; and G is government policy.

When the price and output of agricultural products are fixed and the cost of other production factors is fixed, farmers' profit from agricultural production is as shown in Equation (3):

$$\text{Profit}\pi = P \times Q - C_1 - C_2 \tag{3}$$

where $C_1$ is the input cost of other production factors; $C_2$ is the water cost, which is equal to agricultural water price ($P_a$) multiplied by water consumption amount ($Q_a$). Substituting the demand function of agricultural water consumption into Equation (3) can obtain Equation (4).

$$\text{Profit}\pi = P \times Q - C_1 - C_2 = P \times Q - C_1 - Q(P_a, P, T, E, C, G \cdots) \times P_a \tag{4}$$

According to Equation (4), farmers' profit from agricultural production is related to water consumption amount and water price.

When the water consumption amount is within the water quota, the water price is fixed, the water cost is positively related to the water consumption amount, and the production profit is negatively related to consumption amount. At this time, farmers who pursue the profit maximization tend to reduce the water consumption amount, although this does not significantly affect the output of crops. In this situation, the water consumption amount is lower than the water quota, which realizes agricultural water-saving aims. As the water consumption amount is negatively related to the water price, the higher the water price (only the higher price level, but still the fixed value), the lower the water consumption amount, the greater the water-saving quantity, and the more obvious the water-saving effect. This implies that agricultural water price has an impact on agricultural water-saving behavior, and there is a negative correlation between them.

When the water consumption amount exceeds the water quota, the water price is positively related to consumption amount: the higher the water consumption amount, the higher the water price. At this time, there is still a positive correlation between the water cost and water consumption amount. Farmers who pursue profit maximization tend to reduce the water consumption amount, while this does not significantly affect the crop yield. Similarly, the agricultural water price has an impact on agricultural water-saving behavior, and there is also a negative correlation between them.

It should be noted that in the actual production process, farmers' water consumption behavior is complex and affected by multiple factors. In addition to the price, it is also affected by agricultural water-saving awareness, motivation, education, planting scale, planting income, water resource endowment, agricultural water-saving facilities, agricultural water-saving technology, irrigation services, government policies, and so on. The theoretical analysis shows that farmers' agricultural water-saving behavior in the process of agricultural production will be affected by individual, family, production, water price and water fees, and water-saving cognitive characteristics, as well as other factors.

*2.2. Hypotheses*

Combined with the above theoretical analysis, based on the existing research [9–28], taking farmers as the research object, the research hypotheses are put forward for the factors affecting farmers' agricultural water-saving behavior:

**H1.** *Individual characteristics (e.g., age, gender, education level, etc.), have a significant impact on farmers' agricultural water-saving behavior;*

**H2.** *Production characteristics (e.g., planting area, planting revenue, etc.), have a significant impact on farmers' agricultural water-saving behavior;*

**H3.** *Water resource characteristics (e.g., water resource endowment, irrigation water dependence, satisfaction of irrigation service, etc.), have a significant impact on farmers' agricultural water-saving behavior;*

**H4.** *Water resource management characteristics (e.g., water fees collecting, agricultural water-saving policies publicity, etc.) have a significant impact on farmers' agricultural water-saving behavior;*

Furthermore, other factors have a significant impact on farmers' agricultural water-saving behavior, but the impact degree and direction are different. It is important to verify whether collecting water fees has a significant impact on farmers' water-saving behavior.

**3. Methods**

A questionnaire was used to determine farmers' agricultural water-saving behavior. They were asked "Under the condition of climate change, to cope with seasonal drought and water shortage, have you ever (i) made water-saving improvement for irrigation facilities, (ii) changed crop planting structure, (iii) adopted agricultural water-saving technology?" If they answered yes to one of the above three responses, it is assumed that agricultural water-saving behavior has occurred. Agricultural water-saving behavior has either occurred or not occurred, and a binary choice logistic model was therefore used to explore the determinants [29].

Combined with the earlier description, this study develops the following binary choice logistic model using Equation (5).

$$P = F(Y = 1|X_i) = \frac{1}{1 + e^{-Y}} \tag{5}$$

where $Y = 1$ when agricultural water-saving behavior occurred, and $Y = 0$ when it did not; P presents the probability of agricultural water-saving behavior occurring during agricultural production activity; $X_i$ is actors which affect farmers' agricultural water-saving behavior.

In Equation (5), Y is a linear combination of variables $X_i$ ($i = 1, 2, \dots 15$), as follows:

$$Y = \alpha + \beta_1 X_1 + \beta_2 X_2 + \beta_3 X_3 + \dots + \beta_i X_i + \mu \tag{6}$$

where $\alpha$ is the intercept parameter; $\beta_i$ ($i = 1, 2, \dots 15$) are regression coefficients; and $\mu$ is the random interference. If $\beta_i$ is positive, variable $X_i$ positively affects farmers' water-saving behavior, and negatively otherwise.

To obtain the odds ratio, Equation (5) is logit transferred to the following equation:

$$Ln\left(\frac{P}{1-P}\right) = \alpha + \beta_1 X_1 + \beta_2 X_2 + \beta_3 X_3 + \dots + \beta_i X_i + \mu \tag{7}$$

According to Equation (7), if $X_i$ changes one unit, the odds ratio changes $e^{\beta_i}$, while other variables remain constant, so it is easy to compare the contribution of variables to farmers' water-saving behavior.

## 4. Variables and Data Collection

### 4.1. Variables

Table 1 presents the definitions of each variable. The dependent variable, farmers' agricultural water-saving behavior, is a binary variable set equal to 0 if agricultural water-saving behavior occurred, and 1 otherwise.

**Table 1.** Variables and definitions.

| Variables | Definitions |
|---|---|
| Dependent variable | |
| Farmers' water-saving behavior (Y) | A binary variable set equal to 0 if agricultural water-saving behavior occurred, and 1 otherwise |
| Independent variables | |
| Farmers' demographics characteristics | |
| Age of household head ($X_1$) | The true age of household head (years) |
| Gender ($X_2$) | 0 = Male, 1 = Female |
| Education ($X_3$) | 1 = Illiterate; 2 = Primary; 3 = Junior; 4 = High school; 5 = College |
| Rural post ($X_4$) | 0 = Not village cadres, 1 = Village cadres |
| Household characteristics | |
| Main income source of family ($X_5$) | 1 = Planting; 2 = Aquaculture; 3 = Non-agricultural employment |
| Agricultural income share ($X_6$) | The ratio of agricultural income to total income (%) |
| Production characteristics | |
| Farm size ($X_7$) | 1 = Less than 0.1 ha; 2 = Between 0.1 and 0.5 ha; 3 = More than 0.5 ha |
| Planting season ($X_8$) | The number of planting seasons (times) in one year |
| Proportion of water expenditure ($X_9$) | The ratio of agricultural water fees to total production cost (%) |
| Water resource dependence ($X_{10}$) | 1 = Very small; 2 = Relatively small; 3 = General; 4 = Relatively large; 5 = Very large |
| Water management characteristics | |
| Water resources endowment ($X_{11}$) | 0 = Rich, 1 = Lack |
| Water-saving policy publicity ($X_{12}$) | 0 = No, 1 = Yes |
| Water fees collecting ($X_{13}$) | 0 = No, 1 = Yes |
| Water using characteristics | |
| Agricultural water service satisfaction ($X_{14}$) | 1 = Very dissatisfied; 2 = Relatively dissatisfied; 3 = General; 4 = Relatively satisfied; 5 = Very satisfied |
| Water usage disputes ($X_{15}$) | 0 = No, 1 = Yes |

### 4.2. Data Collection

South China is dominated by a subtropical humid monsoon climate, with abundant rainfall and water resources. However, the seasonal variation of precipitation is large, mainly concentrated in April and June, while July and September are dominated by subtropical highs. Except for local thunderstorms, rainfall is rare, summer water logging and autumn drought are common in most areas, and there are seasonal droughts and water shortages.

A questionnaire was compiled according to the actual situation of agricultural water use in the seasonal water shortage areas in South China, and the final questionnaire was obtained after modification. From October 2018 to June 2019, the research group conducted a questionnaire survey on farmers in some areas of Jiangxi, Hunan, and Hubei provinces (see Figure 2). The survey subjects were mainly rice and vegetable growers. The survey was conducted by random sampling in groups, and the administrative villages were randomly selected from the above provinces.

The preliminary questionnaire was designed with the help of experts and a pre-survey of 50 farmers was conducted to check its rationality and feasibility. The final questionnaire was derived after addressing problems found in the pre-survey. A total of ten surveyors

were trained before the final survey, and the formal survey was conducted in the form of a face-to-face interview. Of the 813 participants, 153 did not provide sufficient information, and the valid sample size was 660 (see Table 2). The excluded information did not differ from that adopted for the analysis, and there was no sample selection bias as the samples were randomly sampled [30].

**Table 2.** Number of participants.

|                | Total              | Invalid          | Valid           |
| -------------- | ------------------ | ---------------- | --------------- |
| Proportion (%) | 813<br>100.00%     | 153<br>18.82%    | 660<br>81.18%   |

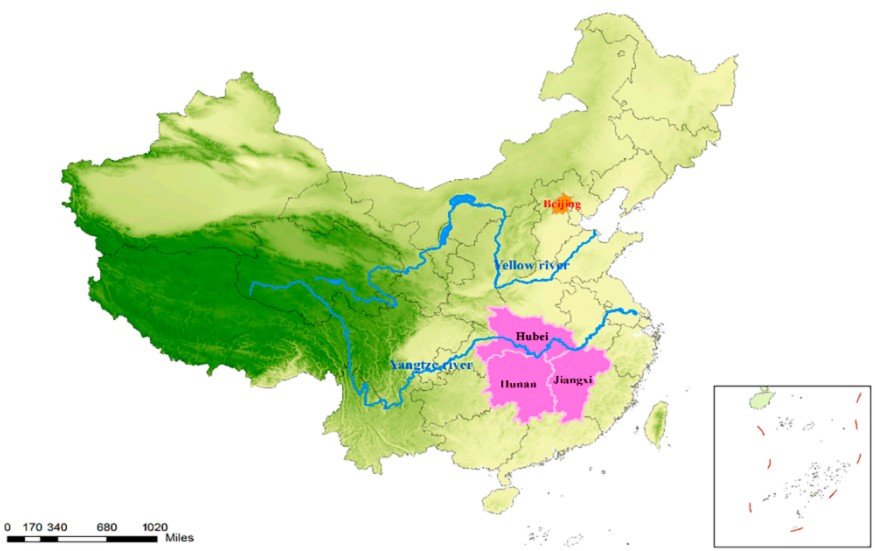

**Figure 2.** The research area.

*4.3. Descriptive Analysis*

4.3.1. Descriptive Analysis of Dependent Variables

In accordance with previous studies, farmers' agricultural water-saving behavior was measured in terms of their water usage behavior. If farmers have made water-saving improvements for irrigation facilities ($B_1$), changed crop planting structure ($B_2$) (e.g., from rice to soybean planting), or adopted agricultural water-saving technology ($B_3$) (e.g., low-pressure pipeline irrigation instead of canal irrigation), agricultural water-saving behavior is considered to have occurred. As long as one or more kinds of the aforementioned behaviors occurred, it is considered to be agricultural water-saving behavior. Survey data showed that 178 (accounting for 26.97%) of the participants engaged in agricultural water-saving, which means that the proportion of agricultural water-saving is low in the survey areas. Table 3 reports farmers' agricultural water-saving behavior. Of the three aforementioned behaviors, 112 farmers have changed their crop planting structure ($B_2$), which is the most common behavior, while fewer farmers engaged in the other two kinds of behaviors. The reason may be that changing crop planting structure is easy to operate, low cost, and does not need technical guidance, so is adopted by more farmers in the case of water shortage. This suggests that the government can encourage farmers in water shortage areas to cope with a water shortage by changing their crop planting structure. Specifically, a total of 156 farmers engaged in only one of the above behaviors. For all of the above three behaviors, 0 engaged in all 3; 13 engaged in $B_1$ and $B_2$; 3 in $B_1$ and $B_3$; and 6 in $B_2$ and $B_3$ (here, $B_1$, $B_2$, and $B_3$ represent the first, the second, and the third types of agricultural water-saving behavior stipulated above, respectively.).

**Table 3.** Agricultural water-saving behavior of farmers.

|  | $B_1$ | $B_2$ | $B_3$ | Water-Saving |
|---|---|---|---|---|
| NO (Y = 0) | 609 | 548 | 626 | 482 |
| YES (Y = 1) | 51 | 112 | 34 | 178 |

### 4.3.2. Characteristics of Agricultural Water Fees

For a long time, it has been the obligation of farmers to pay for water. However, since the implementation of the agricultural tax relief and the state's granting of farm subsidies and other policies to reduce the burden of farmers' production, agricultural water fees have not been collected from farmers in the vast majority of South China but paid by local finance through transfer. In the survey sample, fees were collected from farmers in some areas but not in others. The specific analysis is as follows.

Among the 660 samples, 101 farmers paid agricultural water fees, accounting for 15.30%, and the proportion of water fees to production costs was between 0.89–12.75%, with an average of 5.72%. Among the 101 farmers, 78 had engaged in agricultural water-saving behavior, accounting for 77.23%, while among the farmers who did not pay agricultural water fees, only 100 had agricultural water-saving behavior, accounting for 17.89%. Furthermore, among the 178 farmers who had agricultural water-saving behavior, 78 paid agricultural water fees, accounting for 43.82%. According to the survey data, the proportion of agricultural water charges in the South's seasonal water shortage area is relatively low. In addition, the survey data also revealed that the ratio of agricultural water-saving behavior in the group who paid fees is higher than that of the non-collecting group, as shown in Figure 3. This may mean that collecting agricultural water fees helps to promote agricultural water-saving.

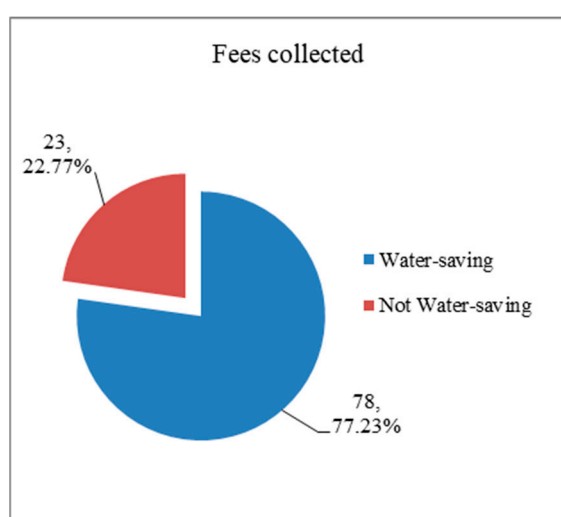
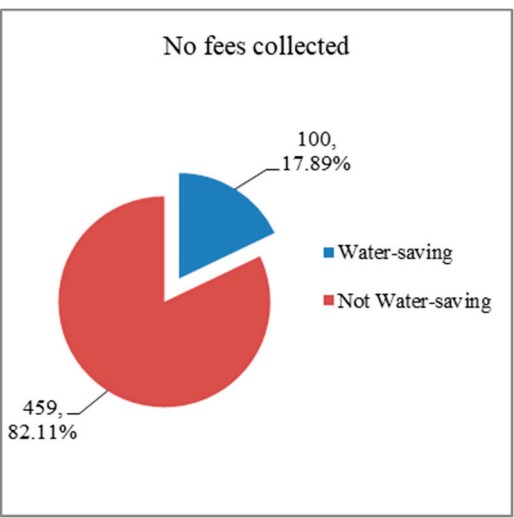

**Figure 3.** Comparison of agricultural water-saving behavior between two groups.

### 4.3.3. Descriptive Analysis of Independent Variables

The sample was divided into two groups according to farmers' agricultural water-saving behavior, and the *t*-test and Chi-square test were used to identify whether variables have significant differences. Table 4 presents the differences in the characteristics of the "Occurred" and the "Not occurred" groups with their $\chi^2$-values and t-values, which indicate that some variables, such as gender, planting season, proportion of water expenditure, water resource dependence, water-saving policy publicity, water fees collecting, and agricultural water service satisfaction are significant. The differences in the mean characteristics between the two groups indicate that these factors may impact farmers' agricultural water-saving behavior.

**Table 4.** Variable differences between "Occurred" and "Not occurred" groups.

| Variables | Units | Occurred (n = 178) | Not Occurred (n = 482) | $\chi^2$/t-Test Values |
|---|---|---|---|---|
| Age of household head ($X_1$) ($\bar{x} \pm s$) | Years | 49.12 ± 11.50 | 49.95 ± 11.23 | 0.842 |
| Gender ($X_2$) (Male)(%) | | 126 (70.8) | 390 (80.9) | 7.815 *** |
| Education ($X_3$) N(%) | | | | |
| Illiterate | | 27 (15.2) | 72 (14.9) | |
| Primary | | 69 (38.7) | 206 (42.7) | 2.427 |
| Junior | | 58 (32.6) | 129 (26.8) | |
| High school | | 14 (7.9) | 44 (9.2) | |
| College | | 10 (5.6) | 31 (6.4) | |
| Rural post ($X_4$) (Not village cadres)(%) | | 144 (80.9) | 410 (85.1) | 1.671 |
| Main income source of family ($X_5$) N(%) | | | | |
| Planting | | 72 (40.4) | 198 (41.1) | |
| Aquaculture | | 8 (4.5) | 11 (2.3) | 2.280 |
| Non-agricultural employment | | 98 (55.1) | 273 (56.6) | |
| Agricultural income share ($X_6$) ($\bar{x} \pm s$) | % | 35.25 ± 27.85 | 35.12 ± 28.89 | −0.042 |
| Farm size ($X_7$) N(%) | | | | |
| Less than 0.1 ha | | 16 (9.0) | 42 (8.7) | |
| Between 0.1 and 0.5 ha | | 103 (57.9) | 269 (55.8) | 0.311 |
| More than 0.5 ha | | 59 (33.1) | 171 (35.5) | |
| Planting season ($X_8$) N(%) | | | | |
| One | | 22 (12.4) | 134 (27.8) | |
| Two | | 142 (79.8) | 308 (63.9) | 18.074 *** |
| Three | | 10 (5.6) | 30 (6.2) | |
| Four | | 4 (2.2) | 10 (2.1) | |
| Proportion of water expenditure ($X_9$) ($\bar{x} \pm s$) | % | 2.76 ± 4.04 | 0.25 ± 1.28 | −7.898 *** |
| Water resource dependence ($X_{10}$) N(%) | | | | |
| Very small | | 1 (0.6) | 46 (9.6) | |
| Relatively small | | 19 (10.7) | 82 (17.0) | |
| General | | 33 (18.5) | 233 (48.3) | 121.429 *** |
| Relatively large | | 104 (58.4) | 90 (18.7) | |
| Very large | | 21 (11.8) | 31 (6.4) | |
| Water resource endowment ($X_{11}$) (Rich)(%) | | 135 (75.8) | 377 (78.2) | 0.421 |
| Water-saving policy publicity ($X_{12}$) (No)(%) | | 97 (54.5) | 369 (76.6) | 30.486 *** |
| Water fees collecting ($X_{13}$) (No)(%) | | 100 (56.2) | 459 (95.2) | 152.927 *** |
| Agricultural water service satisfaction ($X_{14}$) N(%) | | | | |
| Very dissatisfied | | 3 (1.7) | 39 (8.1) | |
| Relatively dissatisfied | | 14 (7.9) | 48 (10.0) | |
| General | | 75 (42.1) | 289 (60.0) | 79.991 ** |
| Relatively satisfied | | 55 (30.8) | 99 (20.4) | |
| Very satisfied | | 31 (17.4) | 7 (1.5) | |
| Water usage disputes ($X_{15}$) (No) (%) | | 94 (52.8) | 326 (67.6) | 12.348 ** |

Note: *** and ** denote significance at 1% and 5%levels, respectively.

### 4.3.4. Correlation Analysis

Table 5 reports the summary statistics of these variables, to identify possible determinants of farmers' agricultural water-saving behavior by using the Pearson correlation analysis and Kendall correlation analysis. These correlation coefficients, which vary from −0.033 to 0.481, between the agricultural water-saving behavior and each factor are reported in the fifth column of the table. The results of the correlation analysis show that gender, planting season, proportion of water expenditure, water resource dependence, water-saving policy publicity, water fees collecting, agricultural water service satisfaction, and water usage disputes are significant correlated with farmers' agricultural water-saving behavior. The water fees collecting ($X_{13}$) variable has the largest correlation (0.481), suggesting a stronger positive effect on farmers' agricultural water-saving behavior than other factors, which means that such behavior can be adjusted by collecting agricultural water fees.

**Table 5.** Summary of statistics.

| | Units | Mean | Standard Deviation | Pearson/Kendall Correlation Coefficients |
|---|---|---|---|---|
| $Y$ (Farmers' agricultural water-saving behavior) | - | 0.27 | 0.444 | - |
| Age of household head ($X_1$) | Years | 49.73 | 11.295 | −0.033 |
| Gender ($X_2$) | - | 0.22 | 0.413 | 0.109 ** |
| Education ($X_3$) | - | 2.50 | 1.049 | 0.012 |
| Rural post ($X_4$) | - | 0.16 | 0.367 | 0.050 |
| Main income source of family ($X_5$) | - | 2.15 | 0.974 | −0.006 |
| Agricultural income share ($X_6$) | % | 35.15 | 28.59 | 0.002 |
| Farm size ($X_7$) | - | 2.26 | 0.607 | −0.019 |
| Planting season ($X_8$) | - | 1.86 | 0.599 | 0.129 ** |
| Proportion of water expenditure ($X_9$) | % | 0.917 | 2.59 | 0.428 ** |
| Water resource dependence ($X_{10}$) | - | 3.16 | 1.102 | 0.326 ** |
| Water resource endowment ($X_{11}$) | - | 0.22 | 0.417 | 0.025 |
| Water-saving policy publicity ($X_{12}$) | - | 0.29 | 0.456 | 0.215 ** |
| Water fees collecting ($X_{13}$) | - | 0.15 | 0.360 | 0.481 *** |
| Agricultural water service satisfaction ($X_{14}$) | - | 3.13 | 0.893 | 0.252 ** |
| Water usage disputes ($X_{15}$) | - | 0.36 | 0.481 | 0.137 ** |

Note: *** and ** denote significance at 1% and 5% levels, respectively.

## 5. Results and Discussion

### 5.1. Estimation Results of Econometric Models

The SPSS20.0 software is used for the regression analysis. First, all variables are included in the model, and the Forward LR regression analysis method is used to obtain the model regression results, which are shown in Table 6 (only the significant factors are listed in the table) [31,32]. As to categorical variables, in order to obtain accurate regression results, using the categorical function of SPSS software, the multi-category variables are transformed into dummy variables, the reference groups and control groups are defined, and the parameters of the different levels of them are also estimated. Combined with the model goodness of fit test reference index, the Chi-square significance level of the model is 0.000, indicating that the estimation results of the binary logistic model are relatively ideal as a whole.

**Table 6.** Estimates of determinants of farmers' agricultural water-saving behavior (logistic model).

| Variables | B | Std. Err. | Wals | Sig. | Exp(B) |
|---|---|---|---|---|---|
| Water fees collecting ($X_{13}$) | 4.850 | 0.500 | 94.158 | 0.000 | 127.761 |
| Water resource dependence ($X_{10}$) | | | 76.002 | 0.000 | |
| Very small (reference group) | | | | | |
| Relatively small | 3.966 | 1.412 | 7.884 | 0.005 | 52.760 |
| General | 3.437 | 1.370 | 6.291 | 0.012 | 31.097 |
| Relatively large | 3.913 | 1.432 | 7.471 | 0.000 | 50.071 |
| Very large | 4.044 | 1.805 | 13.891 | 0.000 | 57.058 |
| Agricultural water service satisfaction ($X_{14}$) | | | 9.200 | 0.000 | |
| Very dissatisfied (reference group) | | | | | |
| Relatively dissatisfied | 3.218 | 1.092 | 8.694 | 0.003 | 24.987 |
| General | 2.702 | 1.027 | 6.925 | 0.008 | 14.905 |
| Relatively satisfied | 3.522 | 1.047 | 11.306 | 0.001 | 33.848 |
| Very satisfied | 4.134 | 1.415 | 8.542 | 0.003 | 62.453 |
| Farm size ($X_7$) | | | 8.796 | 0.035 | |
| Less than 0.1 ha (reference group) | | | | | |
| Between 0.1 and 0.5 ha | −0.404 | 0.932 | 0.188 | 0.032 | 0.668 |
| More than 0.5 ha | −1.050 | 1.093 | 0.923 | 0.009 | 0.350 |
| Water-saving policy publicity ($X_{12}$) | 0.392 | 0.302 | 1.688 | 0.022 | 1.480 |
| Age of household head ($X_1$) | −0.023 | 0.011 | 4.314 | 0.038 | 0.977 |
| Constant | −9.373 | 1.925 | 23.704 | 0.000 | |

Note: −2 Log likelihood = 373.092, LR chi$^2$ (10) = 358.221, Cox and Snell R$^2$ = 0.433, Nagelkerke R$^2$ = 0.631, Prob. > chi$^2$ = 0.0000.

As can be seen in Table 6, water fees collecting, water resource dependence, agricultural water service satisfaction, farm size, water-saving policy publicity, and age of household head are significant determinants influencing farmers' agricultural water-saving behavior. Water fees collecting is positively correlated with farmers' agricultural water-saving behavior, implying that collecting water fees can affect farmers' agricultural water-saving behavior, which also means that collecting water fees can indeed promote agricultural water-saving. These variables are explained below.

(1) Water fees collecting: The regression coefficient is positive and significant at the 1% level, implying that compared with not collecting water fees from farmers, it is more conducive to promoting farmers' water-saving. The reason may be that collecting agricultural water fees from farmers can enhance their water commodity awareness. Along with the strengthening of publicity of water-saving policies in recent years, farmers' water-saving awareness has gradually enhanced, and under the background of the seasonal water shortage, farmers tend to agriculturally take water-saving measures, and then promote agricultural water-saving.

(2) Water resource dependence: Compared with the reference group (Very small), the regression coefficient of control groups is positive and significant at the 1% level, implying that the higher the dependence of agricultural production on water resources, the higher the probability of farmers engaging in agricultural water-saving behavior. When farmers grow crops, their dependence on water resources is related to the number of planting seasons, types of crops, planting time, and so on. For example, when they plant rice only one season per year, the sowing, growth, and maturity of rice are all in the wet season, so the dependence of agricultural production on water resources is low. However, some vegetables planted in autumn depend heavily on water resources. Because the sowing time is in the seasonal water shortage period in South China, farmers tend to save water to ensure the water demand of vegetable production. In addition, in water shortage areas, farmers will change to grow low water consumption crops such as soybean and corn in the original rice planting area.

(3) Agricultural water service satisfaction: Compared with the reference group (Very dissatisfied), the regression coefficient of control groups is positive and significant at the 1% level, implying that the higher the satisfaction with the agricultural irrigation service, the greater the possibility of agricultural water-saving behavior. The reason may be that, in the water shortage area, the water supply and consumption management is strict and a uniform dispatching operation is carried out. Furthermore, the water resource management department scientifically allocates water resources according to the law of crop growth water demand, and guide farmers to use water reasonably. Hence, the more satisfied the farmers are with the irrigation service, the more obedient they will be regarding water management, and use water according to the law of crop water demand to reduce the waste of water resources and realize agricultural water-saving.

(4) Farm size: The regression coefficient is negative and significant at the 5% level, implying that the larger the farm size, the lower the possibility of agricultural water-saving behavior. For the farmers with a large farm size, the purpose of planting crops is to obtain more economic benefits, while agricultural water-saving may not only reduce the yield of crops and reduce the benefits, but also increase the production investment (water-saving transformation), so farmers tend to not save water. Furthermore, the larger the farm size, the more difficult the water management, and the more difficult it is to adopt agricultural water-saving measures.

(5) Water-saving policy publicity: The regression coefficient is positive and significant at the 5% level, implying that the publicity of the agricultural water-saving policy is also helpful to promote agricultural water-saving. The reason may be that there are abundant water resources in South China, and farmers generally lack water-saving awareness. By publicizing agricultural water-saving policies, farmers' water-saving awareness can be enhanced, which also affects their water use behavior and guides farmers to reduce water waste and adopt appropriate agricultural water-saving measures.

(6) Age of household head: The regression coefficient is negative and significant at the 5% level, implying that the older the farmer, the lower the possibility of agricultural water-saving behavior. The reason may be that the older farmers are accustomed to traditional agricultural production methods, and they have no habit of or experience in agricultural water-saving, while the younger farmers are more inclined to accept new things and are willing to respond to the government's call for agricultural water-saving.

## 5.2. Discussion

This study analyzes farmers' agricultural water-saving behavior and its determinants in southern seasonal water shortage areas in China. The results indicate that collecting agricultural water fees can really promote agricultural water-saving in southern China with seasonal water shortages. From the above analysis, this paper identifies several significant determinants that affect a farmer's agricultural water-saving behavior. The results contribute to the literature in two aspects: one is this paper defines agricultural water-saving behavior and three main kinds in southern China through an investigation; another one is revealing that collecting agricultural water fees can really promote agricultural water-saving. Compared to the previous research [18–22], these studies are based on the agricultural water use collecting fees, discussing the impact of raising agricultural water prices [19,20], providing water price explicit subsidies [21,22], and charging water by volume [22] on agricultural water-saving, and the results showed that the above factors affect agricultural water-saving. However, this study focuses on verifying whether collecting water fees can really promote agricultural water-saving in a seasonal water shortage area in South China, with the research object also including the area not collecting water fee, and charging by area can also promote agricultural water-saving; this is different from previous studies [22].

While this study has verified that collecting water fees can really promote agricultural water-saving in a seasonal water shortage area in South China, it has some limitations. The survey does not include all seasonal water shortage areas in South China, which reduces the representativeness of data and the validity of the research conclusions. Meanwhile, this study does not consider other factors such as the water fee levied by volume, farmers' water-saving motivation, water-saving awareness, and agricultural water-saving incentive policy, which may also be significant factors. In addition, more types of water use behavior, not just including the three types investigated herein need to be examined. Furthermore, the agricultural water-saving behavior in South China may have been underestimated. With the implementation of the Rural Revitalization Strategy and the rapid development of agricultural modernization, the pattern of agricultural business subjects is changing, and new agricultural business subjects such as farmers' professional cooperatives, family farms, and large professional households are gradually increasing, which will become the leading force in the construction of modern agriculture in China, and its agricultural water-saving behavior is also worth exploring. This study does not discuss the agricultural water-saving behavior of this group, but rather leaves it to future research.

## 6. Conclusions and Policy Implications

### 6.1. Conclusions

Empirical results reveal that water fees collecting, water resource dependence, agricultural water service satisfaction, and water-saving policy publicity are factors that positively influence farmers' agricultural water-saving behavior, while farm size and age of household head negatively influence it. The results indicate that in these regions, the collecting of agricultural water fees is indeed conducive to promoting agricultural water-saving.

Factors that affect farmers' agricultural water-saving behavior are sorted by contribution. Of the above six factors, water fees collecting is the most important, while water-saving policy publicity is relatively less important. This implies that the role of publicity in promoting agricultural water-saving is weaker than that of economic means. This also means that to promote agricultural water-saving, the role of economic means

should be effectively emphasized. Due to the low education degree in China and the lack of water-saving awareness, the publicity of agricultural water-saving policies only works for those farmers with high education and strong water-saving awareness. However, the collection of agricultural water fees directly affects the production cost and planting income of farmers. From an economic perspective, for rational individuals who pursue profit maximization, it will significantly affect their water use behavior and tend to save water. Hence, as collecting such fees is effective, policy makers should commit to collecting them from farmers and strengthen the collection of such fees. It needs to be emphasized that collecting agricultural water fees is a means, not a goal. The government can give specific subsidies and water-saving rewards according to farmers' water-saving situation, so as to achieve the goal of not only deducing farmers' burdens, but also achieving agricultural water-saving.

### 6.2. Policy Implications

First, an agricultural water charges system should be strictly implemented. Based on the fact that collecting agricultural water fees in seasonal water shortage areas of South China is helpful to agricultural water-saving, policies should be formulated to ensure that the leverage of charging can be effectively exerted. Therefore, the local water administrative department should improve the agricultural water charge policies, select reasonable charging methods, and strengthen the agricultural water charge system by collecting agricultural water fees from farmers, to enhance their water commodities awareness, form their internal water-saving awareness, and promote agricultural water-saving. At the same time, to formulate agricultural water-saving reward and compensation policies, the method of charging water fees first and then compensating according to the water-saving situation can be adopted (collect then refund), for example, to ensure that policy implementation does not increase farmers' water cost; accordingly, agricultural water-saving benefits can also be obtained.

Second, water administrative departments should strengthen the irrigation services. The local water administrative department should formulate an annual water supply plan, strengthen water resource management, optimize water resource allocation, and meet the water demand of agricultural production in all stages. At the same time, the administrative department should strengthen the guidance of water use for farmers, reasonably arrange the irrigation time according to the law of water demand for crops, scientifically determine the amount of irrigation water, reduce the waste of water resources, and improve the utilization efficiency of water resources. In addition, the administrative department should also change its functions from management to service functions, improve farmers' satisfaction with irrigation services, and guide more farmers to use water scientifically and consciously save water.

Third, the publicity of the agricultural water-saving policy should be increased. Holding symposiums, posting slogans, news media, printing and distributing informational materials, publicizing the national agricultural water-saving policy to farmers, and introducing the supply and demand situation of water resources in the seasonal water shortage areas of South China will increase farmers' understanding of the necessity and urgency of agricultural water-saving, its purpose and significance, and further enhance their water-saving awareness. Increasing the publicity of local agricultural water-saving policies, especially the agricultural water-saving incentive policies, enhances farmers' internal power to save water, and improves the enthusiasm for water-saving. In addition, it can also introduce simple and practical agricultural water-saving common sense to farmers and guide farmers to carry out agricultural water-saving practices, so that more water-saving common sense and technology can be transformed into water-saving practices.

**Funding:** This manuscript is supported by the Jiangxi Province of the Social Science Planning Project in the 13th five-year plan (19YJ42), the Jiangxi College of Humanities and Social Sciences project (JJ20214).

**Institutional Review Board Statement:** Not applicable.

**Informed Consent Statement:** Not applicable.

**Data Availability Statement:** Data sharing is not applicable to this article.

**Acknowledgments:** The author would like to express heartfelt thanks to the anonymous reviewers for their helpful comments on the manuscript. Meanwhile, the undergraduates should also be thanked for their hard work in the questionnaire survey.

**Conflicts of Interest:** The authors declare no conflict of interest.

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
