# Peer review of "Can Collecting Water Fees Really Promote Agricultural Water-Saving? Evidence from Seasonal Water Shortage Areas in South China"

_sustainability, doi:10.3390/su141912881_

Round 1

Reviewer 1 Report

The manuscript provides useful information for water-saving. The methods are reasonable and the results can be acceptable. While the structure should be revised.

Specifically, 1) the last paragraph (lines 110-114) can be removed; 2) the section "2. Literature Review" should be integrated into "Introduction" or "Discussion"; 3) The Discussion is insufficiency.

In addition, only one author in this study, which is rare in the complex study and investigation for the 660 farmers. What's more, the email "QQ" is informal.

Reviewer 2 Report

In this manuscript, authors used binary logistic regression to identify the key determinants of farmers’ agricultural water-saving behavior. In my opinion it is a relevant theme for discussion and the authors bring some new perspectives for water conservation analysis. 

However, I think the statistical analysis has some problems. The authors need to review all the statistical analyses performed.

The categorical predictors was treated as numerical. Even if they are coded with numbers they should be analyzed as qualitative variables.  In descriptive statistics it is more appropriate to present the absolute and relative frequencies for categorical variables rather than the mean and standard deviation. 

The use of the student t-test is not suitable for comparing two categorical variables, e.g. water-saving behaviour x gender. Spearman's correlation coefficient is also not adequate to evaluate the relationship between two categorical variables.

In the logistic model, the reference levels of categorical predictors were not defined and the parameters of the different levels of them were not estimated. This can affect the results, especially related to predictors with more than two levels, such as Main income source of family

In section 4 there is no references. 

For these reasons, I am inclined to rejecting this paper.

Reviewer 3 Report

  • The Manuscript with titile "Can collecting water fees really promote agricultural water-saving? Evidence from seasonal water shortage areas in South China" is quite interesting for us in particular as an effort to reduce the water consumption in agricultural sector.
  • The introduction is well written and the topic is interesting for the journal's readership.
  • The method is quite exhaustively.
  • The results are very good and reasonable.
  • Lastly, I recommend to accept this manuscript.

Reviewer 4 Report

Research topic is very important, research is properly done and the paper is well structured.

The author should omit mentioning in the text political bodies and names (line 38, line 47), it is sufficient to mention those in references.

In lines 127 - 130 the phrase ".. influencing factors of ......" appears twice. Check whether that is necessary.

In lines 339-341 some strange sings appear as variables. Please check.

Round 2

Reviewer 1 Report

The author has revised the manuscript according to my comments. I feel that the manuscript can be accepted for publication.

Reviewer 2 Report

Pearson's linear correlation coefficient can only be applied between quantitative variables. Authors should review table 5.

In table 6, please include the reference categories for the categorical variables.
